# 3D Segmentation of Intracerebral Hemorrhage in Brain CT Using Enhanced UNet Transformer via Reinforcement Learning

**Sana Ullah**[1]                                    SANAULLAH@KHU.AC.KR
**Gabin Kim**[1]                                     SARAH242438@KHU.AC.KR
**Dongwook Lee**[1]                                  EHDDNR165@KHU.AC.KR
**Hafiz Zahid Tufail**[1]                            ZAHID.TUFAIL@KHU.AC.KR
**Kangsan Lee**[1]                                   KANGSANL@KHU.AC.KR
**Sunghyun Park**[1]                                 BEAU0124@KHU.AC.KR
**Joon Young Kim**[1]                                JOONYOUNG.KIM0525@KHU.AC.KR
**Won Hee Lee**[2]                                   WHLEE@KHU.AC.KR
**Tae-Seong Kim**[*1]                                TSKIM@KHU.AC.KR

[1] *Department of Electronics and Information Convergence Engineering, Kyung Hee University, Yong-In, Korea*

[2] *Department of Software Convergence, Kyung Hee University, Yong-In, Korea*

## Abstract

Intracranial hemorrhage (ICH) segmentation from 3D CT is critical for treatment planning, as ICH can lead to severe neurological deficits and death. While 2D slice-wise convolutional neural networks and transformer-based models have achieved strong performance, 2D approaches may lose inter-slice continuity that is important for volumetric lesions. In contrast, 3D models capture spatial relationships across the volume but still struggle with accurate delineation of small or low-contrast hemorrhages. To address these limitations, we propose a novel segmentation framework enhanced by reinforcement learning. Our approach, named UNETR-PPO, is an integration of the transformer-based UNETR segmentation model with reinforcement learning. We evaluate our method on the HemSeg-200 3D CT ICH dataset and demonstrate improved performance over the baseline models including 3D U-Net, V-Net, SwinUNETR, and UNETR in terms of Dice, IoU, and HD95.

**Keywords:** Intracranial Hemorrhage, 3D Segmentation, PPO, Reinforcement Learning

## 1. Introductions

Intracranial hemorrhage (ICH) accounts for approximately 10-15% of all stroke cases and is associated with a high risk of mortality (Hostettler et al., 2019). Hemorrhage volume can expand rapidly within the first few hours after onset (Qureshi and Palesch, 2011). Therefore, accurate segmentation of ICH from brain CT scans is essential for surgical planning. Deep learning segmentation models have led to substantial progress in automated ICH segmentation (Ironside et al., 2019). In early deep learning approaches, 2D CNN-based segmentation models were explored for ICH segmentation. For instance, AttFocusNet (Peng et al., 2022) were proposed for ICH segmentation. Because these 2D slice-wise models may tend to

---

neglect inter-slice spatial relationships, segmentation commonly resulted in discontinuous segmentation and reduced robustness (Kumar et al., 2024). To overcome the limitations of 2D models, 3D segmentation models have been developed to exploit the full volumetric anatomic context. For instance, DeepBleed was proposed to maintain consistency across slices and capture 3D lesion morphology (Sharrock et al., 2021). Lately, 3D Transformer models such as UNETR and SwinUNETR have been applied for 3D ICH segmentation (Song et al., 2024). However, even these 3D models still struggle with subtle ICH segmentation, often producing poorly defined boundaries and fragmented lesions (Song et al., 2024), indicating the need for further improvements in segmentation accuracy. Recently, deep reinforcement learning (DRL) has emerged in medical image analysis due to its ability to optimize model decisions using task-driven reward functions (Zhou et al., 2021). For instance, RL4Seg proposed in (Judge et al., 2024), formulates segmentation as a policy optimization problem and employs Proximal Policy Optimization (PPO) to fine-tune a pre-trained model for echocardiography segmentation. Their approach surpasses state-of-the-art domain adaptation methods in segmentation accuracy while producing more anatomically consistent segmentation masks. Similarly, PolicySegNet (Patel et al., 2025) integrates PPO with transformer-based architectures for brain tumor segmentation and demonstrates improved performance. These studies suggest that DRL can enhance segmentation accuracy. However, to the best of our knowledge, RL has not yet been applied to ICH segmentation. In this work, we propose UNETR-PPO, an DRL-enhanced 3D segmentation framework that incorporates PPO with a transformer-based UNETR backbone (Hatamizadeh et al., 2022) to improve ICH segmentation.

## 2. Methods

Figure 1 illustrates the proposed UNETR-PPO framework, which integrates UNETR with actor–critic networks from PPO. UNETR (Hatamizadeh et al., 2022) acts as the backbone 3D segmentation model, using a Vision Transformer (ViT) encoder and a U-shaped CNN decoder. Proximal Policy Optimization (PPO) (Schulman et al., 2017) is employed to refine the segmentation through an actor–critic policy-gradient approach.

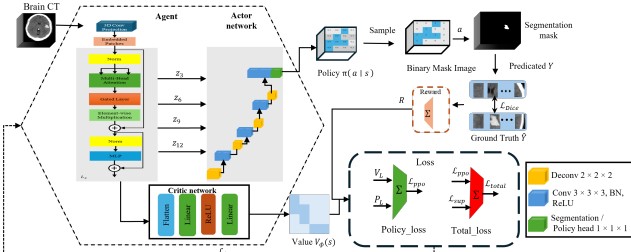

Figure 1: Proposed UNETR-PPO architecture.

**UNETR-PPO.** We formulate ICH segmentation as a single-step episodic RL problem where environment corresponds to an input 3D CT volume. The input volume is passed through a Vision Transformer (ViT) encoder that partitions the volume into non-overlapping 3D patches (i.e., tokens) and uses self-attention to learn global contextual latent representations serving as a state $s$. The actor network follows a U-shape architecture; multi-scale transformer features extracted by ViT are fused into a CNN decoder through skip connections to generate an action, $a$ (i.e., segmentation output) conditioned on the state $s$. The

segmentation model outputs voxel-wise probability segmentation outputs, defining a policy, $\pi(a \mid s)$. In parallel, the critic network operates on deep bottleneck-encoder features and approximates a value, $V_\phi(s)$ which estimates the expected reward at $s$ under the current policy, which serves as a baseline to stabilize policy-gradient learning. We define a reward, $R$ as the Dice similarity between prediction and ground truth (i.e., higher overlap yields higher reward). The reward is calculated according to Equation: $R = 1 - \mathcal{L}_{\text{Dice}}(\hat{Y}, Y)$.

where $\mathcal{L}_{\text{Dice}}(\hat{Y}, Y)$, represents the Dice loss between prediction and ground truth. $\hat{Y}$ denotes the predicted mask. $Y$ denotes the ground-truth mask. The policy $\pi(a \mid s)$ is optimized using the actor-critic PPO loss together with the supervised segmentation loss. **Datasets and Pre-processing.** Experiments were conducted on the publicly available HemSeg-200 3D brain CT dataset (Song et al., 2024), which contains 222 volumetric scans. Background voxels associated with imaging artifacts (e.g., head supports) were removed by cropping non-informative regions from each CT volume, resulting in a curated dataset of 200 volumetric scans.

## 3. Results and Conclusion

Table 1 summarizes the quantitative results achieved through five fold cross validation. UNETR-PPO consistently outperforms all baseline models, achieving the Dice score of 71.36%, IoU of 58.73%, and an HD95 of 34.34. These gain highlights the significance of PPO-driven refinement for segmenting small hemorrhages.

Table 1: Comparative analysis for ICH segmentation of UNETR-PPO against baseline 3D segmentation models.

| Model | Dice (%) | IoU (%) | HD (mm) |
|---|---|---|---|
| U-Net (Çiçek et al., 2016) | 63.32 | 49.76 | 54.42 |
| V-Net (Milletari et al., 2016) | 70.13 | 57.37 | 34.58 |
| SwinUNetr (Hatamizadeh et al., 2021) | 54.78 | 40.93 | 167.11 |
| UNETR (Hatamizadeh et al., 2022) | 59.53 | 45.47 | 103.05 |
| **UNETR-PPO** | **71.36** | **58.73** | **34.24** |

Qualitative comparisons are illustrated in Figure 2 via 3D volumetric and 2D slice-wise visualizations of two patients respectively. In Figure 2, the ground-truth lesions are shown in red, while the predicted segmentation masks in green.

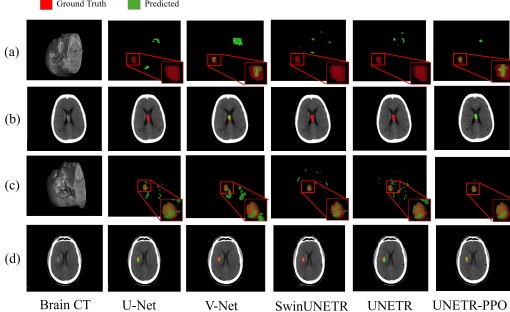

Figure 2: ICH segmentation comparison for two patients: (a,b) Patient A: 3D visualization and 2D CT overlays; (c,d) Patient B: 3D visualization and 2D CT overlays.

**Acknowledgments.** This research was supported by a grant of the Korea Health Technology R&D Project through the Korea Health Industry Development Institute (KHIDI),

funded by the Ministry of Health & Welfare, of Korea (RS-2025-02220492). This work was supported by the Institute of Information & Communications Technology Planning & Evaluation (IITP) grant funded by the Korea government (MSIT) (No. RS-2024-00509257, Global AI Frontier Lab). This work was supported by the IITP (Institute of Information & Communications Technology Planning & Evaluation) - ITRC (Information Technology Research Center) grant funded by the Korea government (Ministry of Science and ICT) ( IITP-2026-RS-2024-00438239).

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
