# OpenReview forum: "3D Segmentation of Intracerebral Hemorrhage in Brain CT Using Enhanced UNet Transformer via Reinforcement Learning"
_MIDL.io/2026/Short_Papers — MIDL 2026 - Short Papers Poster_

### Official Review · Reviewer_W1m2 · 2026-04-24
**Good Paper**

**Rating:** 5
**Confidence:** 4

**Review:**

For a short paper contribution, this paper is well-written and provides the necessary information for acceptance.

**Summary:**

The paper proposes to use Reinforcement learning with a Transformer-based segmentation network for Intracerebral Hemorrhage segmentation. Experiments and results on a publicly available dataset with five-fold cross-validation demonstrate the usefulness of the proposed method compared to other relevant baselines.

**Strengths:**

* Novel use of RL-PPO for Intracerebral Hemorrhage segmentation.
* Really good introduction and literature review for a short paper.
* Good description of the methodology.
* Comparison against other baselines is good.

**Weaknesses:**

* Both Figures are really small. I understand that it might be due to space constraints. But in that case, authors could have put both side-by-side to save some space and make them bigger.
* It would be a good idea to add variance across all 5 folds. This would be useful considering that V-Net and the proposed methods give somewhat similar performance to each other.

**Justification Of Rating:**

The paper is well organized, with relevant baselines and enough contributions for a short paper.

---

### Decision · Program_Chairs · 2026-05-08

Accept (Poster)